# Do non-traumatic stressful life events and ageing negatively impact working memory performance and do they interact to further impair working memory performance?

Denise Wallace[1]*, Nicholas R. Cooper[1], Alejandra Sel[1], Riccardo Russo[1,2]

1 Department of Psychology and Centre for Brain Science, University of Essex, Colchester, Essex, United Kingdom, 2 Department of Behavioral and Brain Sciences, University of Pavia, Pavia, Italy

☯ These authors contributed equally to this work.
* dwallace@essex.ac.uk

**Data Availability Statement:** All relevant data are within the paper and its Supporting Information files.

## Abstract

Stress and normal ageing produce allostatic load, which may lead to difficulties with cognition thereby degrading quality of life. The current study's objective was to assess whether ageing and cumulative stress interact to accelerate cognitive decline. With 60 participants, Marshall et al. found that ageing and cumulative stress interact significantly to impair working memory performance in older adults, suggesting vulnerability to the cumulative effects of life events beyond 60 years old. To replicate and extend this finding, we increased the sample size by conducting 3 independent studies with 156 participants and improved the statistical methods by conducting an iterative Bayesian meta-analysis with Bayes factors. Bayes factors deliver a more comprehensive result because they provide evidence for either the null hypothesis (H0), the alternative hypothesis (H1) or for neither hypothesis due to evidence not being sufficiently sensitive. Young (18–35 yrs) and older (60–85 yrs) healthy adults were categorised as high or low stress based on their life events score derived from the Life Events Scale for Students or Social Readjustment Rating Scale, respectively. We measured accuracy and reaction time on a 2-back working memory task to provide: a) Bayes factors and b) Bayesian meta-analysis, which iteratively added each study's effect sizes to evaluate the overall strength of evidence that ageing, cumulative stress and/or the combination of the two detrimentally affect working memory performance. Using a larger sample (N = 156 vs. N = 60) and a more powerful statistical approach, we did not replicate the robust age by cumulative stress interaction effect found by Marshall et al.. The effects of ageing and cumulative stress also fell within the anecdotal range (⅓<BF<3). We therefore conclude that there was inconclusive statistical evidence, as measured with a life events scale, that ageing and cumulative life stress interact to accelerate cognitive decline.

**Funding:** The author(s) received no specific funding for this work.

**Competing interests:** The authors have declared that no competing interests exist.

## Introduction

Both stress and natural ageing independently produce allostatic load within the brain and body. Allostatic load is the accumulated wear and tear caused by the repeated attempts to adapt to change [1,2]. Within the brain, allostatic load translates into structural changes over time which include retracted dendrites and spine density, spine loss [3] and reduced hippocampal neurogenesis [4–10]. Consequently, cognitive higher functioning can deteriorate resulting in difficulties with memory and attention. These executive functions drive decision-making and learning and, without their optimal function, quality of life suffers especially with advancing age.

Two areas of the brain that are particularly important to cognitive function and most vulnerable to the effects of allostatic load due to ageing and stress [11,12] are the hippocampus and the prefrontal cortex (PFC). For example, ageing has been linked to reduced white matter [13] and myelination integrity [14]. This deterioration starts in the neocortex [15,16] along with generally lower levels of neurotransmitters [17, and see 18 for general review] and consequently fewer receptors, leading to less efficient neural connections [19,20]. These structural and chemical changes then manifest in poorer behavioural outcomes. Studies of healthy ageing that compare young with older adults have shown that young adults outperform older adults on a range of tasks measuring processing speed, working memory and episodic memory in particular. For example, Vasquez and colleagues [21] administered executive tasks for switching, inhibition, fluency, problem solving and working memory in young and older adults. They found that older adults' executive control was poorer than that of young adults as evidenced by longer, more variable reaction time, poorer accuracy and greater variability in performance across tasks within subjects; these effects were found to be linear with age with the most elderly participants (75–85 yrs) performing worst. Ageing is also associated with endocrine changes, such as increased diurnal cortisol levels, which have been associated with hippocampal volume atrophy. The hippocampus, PFC and amygdala contain naturally high concentrations of glucocorticoid receptors [22–24] which allows for enhanced malleability, facilitating dynamic and flexible responsivity to the environment. However, these receptors can be flooded by glucocorticoids in response to perceived stress. Both animal and human studies have demonstrated that perceived environmental stressors accelerate the ageing process within the brain, which can moderate how well we perform mental tasks [25–28]. For instance, in a population-based longitudinal study of adults aged 65 years and older (n = 6207), increased levels of perceived stress predicted poorer cognitive performance and a faster rate of cognitive decline [29]. The ability of glucocorticoids to modulate memory function is well-documented [30,31] and, taken to the extreme, can be toxic to neurons causing permanent structural damage [32].

To investigate the combined allostatic load of ageing and stress, previous research has focused on chronic stress. Souza-Talarico and colleagues [33] concluded in their review that chronic stress exposure, in the context of ageing, shows similar markers to Alzheimer's Disease regarding dendritic atrophy and oxidative stress. Another approach has been to investigate the cumulative effect of life events stress. Holmes and Rahe [34] found that the same cluster of life events typically preceded illness onset, which led them to develop and validate the Social Readjustment Rating Scale (SRRS). It was subsequently used to demonstrate predictive validity of illness onset in clinical settings [35]. Holmes and Rahe [34] argued that these life events disrupted the status quo of day-to-day life and required a certain amount of adjustment, which they operationalised as a numerical weight ranging in value from 0 to 100 based on the averaged weightings assigned by a sample of male and female adult raters. The SRRS asks responders to indicate which of 43 events they have experienced over the last 12 months. The events

comprise a variety of life stressors that can vary in severity and rely on how the individual appraises the event (e.g. 'change to a different line of work', 'death of a close friend', 'outstanding personal achievement'). The sum of the selected (weighted) events produces a 'life events score', which can then index the burden of cumulative stress over 12 months. Marshall and colleagues extended the original SRRS by asking participants to report which events they had experienced over the course of their lives. The sum of the selected events produced a life events score representing cumulative life stress experienced over their entire lives. For the young adult sample, Marshall et al. used the Life Events Stress Scale, which is based on the SRRS and was developed in the same way, using weighted scores. Using this approach in a series of cross-sectional studies, Marshall and colleagues compared young and older adults with varying levels of cumulative stress on a range of cognitive tasks paired with EEG [36–38]. Their key finding was that older adults who reported higher levels of cumulative stress performed less well in working memory, inhibitory control and spatial discrimination tasks compared to their lower stress counterparts and the young-adult sample. Given the consistency of this finding across tasks, the deleterious impact of stress on cognition does indeed appear to be cumulative and impacts a broad spectrum of executive functions. Furthermore, the accompanying resting- and active-state EEG data revealed changes in oscillatory dynamics, such as power and synchronisation of theta [37] and alpha frequencies [38], which were associated with deficits in performance and early signs of cognitive decline [39]. Capitalising on these findings, we conducted a study with the aim of replicating the interaction between age and experienced stress found by Marshall et al. [38] and extending this research by developing a neurostimulation-based treatment protocol with the intention of mitigating the impact of stress and ageing on cognitive function.

We chose to replicate the working memory study because it is a reasonable proxy for higher cognitive function [40]. Working memory is a multi-functional system that allows one to hold in mind a small number of elements, for a few seconds, whilst simultaneously manipulating them for some goal-directed purpose such as comprehension, learning and problem-solving [41–43]. It is supported by a broad range of anatomical structures, including the PFC and medial temporal lobe [44–49], which are connected via short- and long-range neural networks [50,51]. These networks are highly vulnerable to the effects of ageing [52–54] and stress [25,26,55,56]. Indeed, working memory impairment is a central component in most neurological and neurodegenerative disorders [57–62].

Briefly, we found a statistically significant ageing effect but no evidence of an interaction between age and cumulative life stress. However, our initial sample size was considerably smaller than Marshall et al.'s [38] (n = 15 vs. n = 60) due to our study design. Interestingly, the findings of a recent longitudinal study by Sussams and colleagues [63] were also incongruent with the hypothesis that accelerated brain ageing follows from higher levels of cumulative stress. They found no evidence for a relationship between an objective life event measure, perceived stress, increased rate of cognitive decline or conversion to dementia by the end of their ≤ 5.5 year longitudinal study. Their elderly sample comprised control (n = 68) and mild cognitive impaired (n = 133] participants, assessed at baseline. They did find, however, that there was an impact of cortisol on cognitive performance, which was present at baseline. Given the incongruence between these findings and those of Marshall et al. [38] and given the implications for research into the impact of life events stress, we conducted 2 further follow-up studies with larger independent samples. These studies also found no evidence of an interaction effect. Thus, the question remains whether repeated stress response activations from stressful life events (i.e. cumulative stress) can accelerate brain ageing, particularly in combination with normal ageing.

We note that our research was conducted around the period of the Covid-19 pandemic, which was likely to have been particularly stressful given the level of disruption and uncertainty at the time. From this perspective, Marshall and colleagues' [38] study was conducted well before the pandemic while our first study (Study 1) was conducted from September 2019 to March 2020, just before the first ever UK Covid-19 lockdown [64]. Our subsequent 2 studies, which were run in parallel (Study 2A and Study 2B), followed in April 2021 just after the easing of full lockdown measures. One would therefore expect stress to be greater in our three studies relative to Marshall's study and, consequently, finding an interaction between ageing and stress may arguably be more likely. However, given that our individual study results did not find evidence of any interaction we believe our results are robust.

The aim of the present study, given the afore-mentioned inconsistencies in findings, was to assess all the data we collected using an iterative Bayesian meta-analysis with Bayes factors as an alternative approach to standard null hypothesis significance testing. The advantage of the Bayes factor, defined as the ratio of the likelihood of the alternative hypothesis (H1) to the likelihood of the null hypothesis (H0), is that it provides a relative indicator of the strength (sensitivity) of evidence for two competing hypotheses irrespective of power [65]. Three conclusions may be drawn from a Bayes factor: evidence for the null hypothesis (H0), evidence for the alternative hypothesis (H1) or evidence for neither hypothesis because the presented evidence is not sufficiently sensitive [65]. By also accounting for the potential lack of sensitivity in the data, we would provide a more comprehensive result to inform future research in this field. Moreover, using a meta-analytic approach provides information about the overall size and consistency of any effect. Applied in the present case, we took the novel step of conducting a Bayesian meta-analysis on the effect sizes of all the individual studies (incorporating Marshall et al.'s study into the prior model) to indicate whether there is a negative impact on working memory (H1) or no such effect (H0) or that there is insufficient evidence for either hypothesis for a) age, b) cumulative stress and c) the interaction of age and cumulative stress.

## Materials and methods

All studies (Study 1, Study 2A and Study 2B) and their procedures were approved by the Science and Health Faculty Ethics Subcommittee 3 of the University of Essex. All procedures were carried out in accordance to the Declaration of Helsinki. In Study 1 participants gave written informed consent. For Studies 2A and 2B, being online only studies, participants had to select 'yes' or 'no' on-screen for each consent statement in lieu of providing written consent before being allowed to proceed to the study. All participants gave informed consent before participating.

### Participants

A total of 173 individuals took part across three replication studies. Within each study, some participants were excluded from analysis for various reasons including not completing the task properly (S1 Appendix provides participation details by study). The total number of participants excluded across the 3 studies was 17, leaving a total analysed sample of 156 individuals. In Study 1, the final analysed sample comprised 19 older (M = 69.1, SD = 6.4, range = 60 to 84, 12 females) and 21 young adults (M = 21.2, SD = 4.3, range = 19 to 34, 16 females). In Study 2A, the final sample comprised 31 young (M = 28.5, SD = 4.0, range = 21 to 34, 13 females) and 27 older participants (M = 64.1, SD = 4.0, range = 60 to 73, 18 females). In Study 2B the final sample comprised 29 young (M = 27.9, SD = 4.9, range = 18 to 35, 22 females) and 29 older participants (M = 64.7, SD = 5.0, range = 60 to 79, 17 females). Median values and inter-

quartile ranges by age are also provided (S1 Table). All participants were right-handed as assessed by the Edinburgh Handedness Inventory.

The first of the 3 studies was conducted in person and the 2 remaining studies were conducted exclusively online. We recruited participants for Study 1 from the local community in Colchester, UK and academic staff and students at the University of Essex (September 2019 to March 2020). All other participants were recruited via Prolific, an online participant recruitment platform (April to May 2021).

We excluded individuals with a history of substance/alcohol abuse as well as those who had: experienced a traumatic childhood event such as sexual/physical abuse; Type 1 diabetes; a severe heart condition; any neurological (e.g. stroke, mild cognitive impairment, Parkinson's Disease, epilepsy) or psychiatric (e.g. depression, anxiety) conditions; a learning difficulty (e.g. dyslexia). We also excluded individuals taking psychoactive medications. In Study 1, participants were screened for suitability prior to attending their first session. Participants who were eligible were then invited to attend the study. For the online Prolific studies, prospective participants were presented with these items as a list prior to signing up for the study and asked not to sign up if they met any of these specified exclusion criteria. For the online studies, we also selected our participant pool based on the above criteria within the Prolific platform, where the options were available, to ensure that we recruited suitable volunteers. Additional checks were embedded within the questionnaire aimed at retrospectively excluding participants who were unsuitable. For example, participants who consumed alcohol within 12 hours of participation were excluded, as were individuals taking prescription medications causing drowsiness. Participants were paid or received student credits. All participants provided informed consent following a description of the tasks and procedures, which were included in an information sheet and again prior to each task.

## Measures

**Measures of demographics, cumulative stress and general well-being.** Participants completed a range of self-report measures. We used all the same indices as Marshall and colleagues [38] for perceived stress and anxiety. In addition to these replicated questionnaires, we asked participants to report their subjective sleep quality and resilience as part of our extension study (details of these statistical outputs are provided in S2 Appendix).

Life Events as a measure of cumulative stress: Cumulative stress was measured as the accumulated effect of experienced stress which accompanied adjustments made to events/changes over the course of participants' lives as set out in Marshall et al. [38]. For example, death of a close friend, taking out a mortgage or changing schools. The Social Readjustment Rating Scale (SRRS) [34] comprises 43 items and was used for the older participants. The Life Events Scale for Students (LESS) [66], comprising 36 items, was administered to the young participants.

For Study 1, the instructions given in both questionnaires were: 'Please indicate which of the following events have occurred in your life. If any event occurred more than once, provide the number of times the event occurred. If the event did not occur, choose zero.' All responses were converted to binary units and then multiplied by the given 'weight' or life change units (LCU) and summed to give a total life events score for each participant. For the LESS, scores ranged from 0–1849 and for the SRRS, scores ranged from 0–1466. For Study 2A and 2B, we asked participants to simply indicate whether each event had occurred, given that we analysed these as a binary variable in Study 1. Participants were therefore simply asked to indicate 'yes' or 'no' whether each item had occurred in their life.

Perceived Stress: We measured current perceived stress with the Perceived Stress Scale-10 (PSS-10) [67,68], which "...measures how unpredictable, uncontrollable and overloaded

respondents find their lives." (p. 43) [69]. The PSS-10 has good internal consistency (Cronbach α > .70) and re-test reliability (> .70). The PSS-10 comprises 10 questions relating to how often certain thoughts and feelings have occurred in the last month on a 5-point Likert Scale. Responses range from 0 ('Never') to 4 ('Very Often'). Six of the questions are negative (1,2,3,6,9,10), representing the subscale 'perceived distress/helplessness', while the positive items, represent the 'perceived coping/self-efficacy' subscale. Perceived stress is measured using the obtained sum total of all items (with positive items being reverse-scored first). Possible score range: 0 to 40 with higher scores indicating a greater level of perceived stress. The PSS-10 has been validated in a wide range of populations including elderly individuals [70].

Sub-clinical Anxiety: State and trait anxiety was measured with the Spielberger State-Trait Anxiety Inventory (STAI) Y Form [71,72], which has been validated across a wide range of populations [72–74] showing good internal consistency (Cronbach α ≥ .70) and test-retest reliability (≥.40 state; .86 trait) [72,75,76]. The state anxiety scale of the STAI comprises 20 statements focused on the intensity of feelings at the present moment. The STAI-S ratings range from 1 ('Not at all') to 4 ('Very much so'). The STAI trait scale comprises 20 statements focused on the frequency of feelings generally. Ratings for the STAI-T range from 1 ('Almost never') to 4 ('Almost always'). Possible scores range from 20 to 80 on each scale (STAI-S, STAI-T). A higher score indicates greater anxiety [72].

To assess working memory, participants completed the 2-back task [77] using the Inquisit platform [78] as in Marshall et al. [38]. In Study 1 and 2A, participants practiced the 1- and 2-back task followed by 2-back experimental trials. Practice sets comprised 20 trials each. In Study 2B, participants received 1-back practice trials followed by 1-back experimental trials and 2-back practice trials followed by 2-back experimental trials. In Study 1, the task was explained verbally in addition to on-screen instructions. Participants confirmed that they understood the task before starting the practice. For online participants (Study 2A and 2B), to compensate for the lack of in-person instruction, participants were shown a detailed demo with instructions and could replay this if they wished. In addition, those scoring below 65% in either 1-back or 2-back practice trials completed an additional set of 20 trials in the respective condition automatically prior to moving on to the experimental trials. The maximum number of practice trials per version was 40 (2 sets of 20). Across all studies, 10 participants repeated the practice trials.

The stimuli presented were Arabic numbers 1–4 (Helvetica) embedded within a 50% random noise grey background. For the 1-back task, participants responded by pressing the space-bar if the current item was the same as the one presented before. For the 2-back task, participants were asked to do the same but matching the current item to the one presented 2 positions before. Each trial was preceded by a blank black screen presented for 200 ms, followed by a randomly selected stimulus slide (number 1, 2, 3 or 4) presented for 500 ms, with an inter-trial interval of 2500 ms. Participants had the full 3000 ms (500 ms +2500 ms) to respond. There were 39 targets and 81 non-targets in total, equally distributed across 3 blocks of 40 trials. Thus, 13 targets and 27 non-targets per block. The blocks were split by two self-paced breaks. There were two measurement indices: reaction time and accuracy.

N-back trials comprise hits, misses, correct rejections and false alarms. We measured percent correct (hits + correct rejections/120 trials *100) and d-prime values for accuracy and reaction time in milliseconds for correct hits. To minimise the statistical impact of response bias, d' values were calculated by block for each subject as follows: The z transformations were derived using the statistical formula NORMSINV(Hit rate)–NORMSINV(False alarm rate) in Microsoft Excel. Perfect scores were adjusted using these formulae: $1 - 1/(2n)$ for perfect hit rate, and $1/(2n)$ for zero false alarm rate, where n was number of total hits and false alarms, respectively [79,80]. Higher d' values represent better accuracy, while a negative d' represents

response confusion and/or response bias [79]. This was true of our sample, as those with a negative d' value had either a high number of misses (11/13 trials) indicating response confusion or false alarms (16/27 trials) indicating response bias. Only blocks with positive d' values were therefore analysed. Note that we will only report percent correct values for accuracy, which were comparable to d-prime values for all analyses. Frequentist statistical outcomes by study for d-prime and percent correct values are given in (S2 Table).

Each participant's reaction time (RT) means and standard deviations were calculated per block for hits. Only RTs within 2.5 standard deviations of the mean were analysed for each participant (details given in Results section) to reduce statistical bias.

## Design and statistical analysis

Prior to any analyses, participants' cumulative stress scores were categorised into 'high' or 'low' cumulative stress groups based on a median split value derived from the respective age-specific experienced stress questionnaires. While this approach is controversial, it is a valid choice provided that the independent variables are uncorrelated [81,82]. Moreover, this method would allow us to evaluate performance differences between high and low levels of cumulative stress with two different life events scales and conduct our intended analyses. The 'low stress' group denoted those who had experienced a lower level of cumulative stress over the course of their lives thus far whilst the 'high stress' group denoted those who had experienced a higher level of cumulative stress over their lives thus far. In Study 1, the median split value for young adults, based on the LESS, was 592 (IQR: 492 to 639.5). For older adults, the value was 913 (IQR: 753 to 1009), based on the SRRS. In Study 2A, the median split LESS value for young adults was 577 (IQR: 357 to 691) and for the older group, the SRRS value was 738 (IQR: 632–845). In Study 2B, the median split values were 516 (IQR: 348–692) and 766 (IQR: 684.5 to 849.5), respectively. Any LESS or SRRS values that fell on the median were allocated to the low stress group, the more conservative approach given our hypothesis. We further provide a supplemental table with each participant's total cumulative stress score for each study, which shows that participants varied in cumulative stress within and between age groups (S3 Table).

Prior to the planned analysis of N-back data we assessed the data for evidence of a speed-accuracy trade-off for hits using Pearson Product Moment Correlation Coefficients [83]. The variables used were percent correct hits and mean RTs for correct hits, therefore a speed-accuracy trade-off would be indicated by a positive correlation. Where a speed-accuracy trade-off was found, percent correct statistically significant results are still reported but conclusions in these cases are based on RT data only as RT data provides a relatively more sensitive representation of any evidence of effect caused by ageing and/or cumulative stress.

Bayes factors (BFs) were calculated by study, based on mean differences for percent correct and reaction time. These statistics were computed for the main effects of age group, stress level, their interaction and the effects of stress level within each age group. For the interaction effects analysis only we calculated mean differences for the harmonic rather than arithmetic mean, given uneven groups.

Following the method provided by Dienes and colleagues [84] using Dienes' calculator http://www.lifesci.sussex.ac.uk/home/Zoltan_Dienes/inference/bayes_normalposterior.swf found at http://www.lifesci.sussex.ac.uk/home/Zoltan_Dienes/inference/Bayes.htm (web link, p. 118) [85], posterior means and standard deviations were used to meta-analyse the strength of the overall evidence for an effect for our three studies. This method is suitable where the series of studies to be meta-analysed are based on the same hypothesis with the same dependent variable [85], as in the present report. We also calculated a BF for each iteration and for

**Table 1. Order of steps for study effect sizes entered into the iterative Bayesian meta-analysis.**

| Iterations | prior values | likelihood values | posterior values |
|---|---|---|---|
| Step 1 | Marshall et al. (2015) [38] mean difference (SEM)[a] | Study 1 mean difference (SEM) | Marshall et al. (2015)* [38] Study 1 mean, SD, 95% credible interval[c] |
| Step 2 | Study 1 posterior mean and SD[b] | Study 2A mean difference (SEM) | Study 1*Study 2A mean, SD, 95% credible interval |
| Step 3 | Study 2A posterior mean and SD | Study 2B mean difference (SEM) | Study 2A*Study 2B mean, SD, 95% credible interval |

[a] SEM = standard error of the mean.

[b] SD = standard deviation.

[c] Upper and Lower values representing the range of credible effect size values. If this range includes zero, H0 is more likely.

Marshall et al.'s [38] reported effect sizes. For our prior model, we assumed a normal distribution given that most values were expected to be within 2 standard deviations of the mean.

We started the iterative meta-analysis with Marshall et al.'s study [38]. Our steps are set out in Table 1, which were as follows: in the first step, the mean difference and the standard error of the mean difference (SEM) for Marshall et al.'s study [38] served as the prior for Study 1 while the likelihood comprised the mean difference and SEM for Study 1. In the second step, the posterior mean and standard deviation calculated from the preceding step served as the prior for Study 2A and the likelihood comprised the mean difference and SEM for Study 2A. In the third and final step, the resulting posterior mean and standard deviation from step 2 served as the prior for Study 2B. Study 2B's mean difference and SEM was the likelihood which provided a posterior mean and standard deviation which provided the effect size for each effect.

To calculate the BF, we used Diene's BF calculator (https://harry-tattan-birch.shinyapps.io/bayes-factor-calculator/). Our steps for BF calculations are set out in Table 2. To calculate a BF for Marshall et al.'s [38] effect size we used an estimated expected effect size as prior for each of our 3 effects under test. We used this approach because aside from Marshall et al.'s work, no previous research has investigated the effects of cumulative life events stress nor how such effects interact with age in this way. Bayesian inference, unlike frequentist methods, views probability as subjective [86,87]. Thus, one may start with a subjective (prior) belief about the credibility of probabilities, which can be mathematically described as a distribution with a probable point estimate. Bayesian inference then incrementally reallocates the credibility of probabilities [86] when new data are introduced. Thus, Bayesian inference provides a method to reach an increasingly more likely outcome [88].

To calculate the BFs for each step of our meta-analysis we used 0.5 of the upper credible interval of the posterior from the previous step as prior, where possible. As stated in Table 2,

**Table 2. Bayes Factor input variables.**

| Iterations | prior value | likelihood values | BF |
|---|---|---|---|
| Step 0 | estimated effect size[a] | Marshall et al. (2015) [38] mean difference (SEM)[b] | Marshall et al., 2015 [38] |
| Step 1 | Marshall et al. (2015) [38] mean difference (SEM) | Study 1 mean difference (SEM) | Study 1 |
| Step 2 | Study 1 posterior 0.5*upper credible interval[c] | Study 2A mean difference (SEM) | Study 2A |
| Step 3 | Study 2A posterior 0.5*upper credible interval | Study 2B mean difference (SEM) | Study 2B |

[a] Based on a reasonable expected maximum difference between groups.

[b] SEM = standard error of the mean.

[c] Upper values of the 95% credible interval represent the maximum likely effect size. Fifty percent of this value represents 1 standard deviation and serves as the prior.

the upper credible interval of the posterior represents the maximum likely effect size and 0.5 represents one standard deviation. For Step 1's BF we used Marshall et al.'s effect size as prior because there was no posterior. The BFs for Steps 2 and 3 were calculated with 0.5 of the upper credible interval of the previous step, as shown in Table 2. The likelihood values were the same as for the meta-analysis.

The final row in Tables 3 to 5 provides the meta-analytically derived evidence of effect for age group, stress level and within-age groups' performance differences due to high vs. low cumulative stress levels. Wetzels and Wagenmaker's [89] classification indicated that BF's values ranging from 3 to 10 represent "substantial evidence for H1", values ranging from 10 to 30 represent "strong evidence for H1" and values ranging from 30 to 100 represent "very strong evidence for H1". Values in the range of <3 to > ⅓ are regarded as "anecdotal (insensitive) evidence for H1" with '1' representing no evidence in either direction. Values ranging from ⅓ and smaller provide evidence of increasing strength for the null. For interest, we also provide a standard meta-analysis for our 3 replications studies in the (S6 Appendix). Robustness checks are calculated alongside the final BF. Briefly, a robust BF is indicated by the extent to which different prior distribution scale factors produce a consistent BF. Scale factors applied to test the robustness of the BF ranged from 3 (which represents the t-distribution at 2 degrees of freedom) to 7 (half-Cauchy distribution, which is equivalent to the t-distribution with 1 degree of freedom) [90,91].

Note that sleep quality and resilience were not included in our analysis because these measures form part of a wider extension project which includes the present replication of Marshall et al.'s [38] study who did not test these variables in their work.

## Procedure

In all studies, participants were told that they would be taking part in a study comprising questionnaires and one or more cognitive tasks, depending on the study, and provided with a comprehensive explanation prior to each measure. Fig 1 provides the basic structure (S4 Appendix provides the full procedure and set of tasks for each study).

## Results

### Biodemographical and self-reported anxiety, stress and resilience outcomes

Table 3A–3C shows biodemographic variables by study. For Study 2A, Table 3B shows a statistically significant difference in STAI-T scores in the older group with low stress (LS) older adults scoring on average -10.90 (SE 4.15) points lower than high stress (HS) older adults: (t (18) -2.654, p = 0.025, bias-corrected and accelerated (BCa) 95% CI: -19.62 to -2.36. There were no other statistically significant differences. For Study 2B, Table 3C shows a statistically significant difference for exercise among older adults only, with a mean difference of -1.44 hrs (SE 0.55) indicating that the HS group spent more time exercising on average than the LS group: t(27) -2.530, p = 0.012, BCa 95% CI: -2.45 to -0.29. No other comparisons were statistically significant. Table 3D provides means and standard errors by age group by stress group for each study and shows no overlap between high and low stress groups within age group for any studies.

### N-back task

Data removed following RT trimming (trials > 2.5 SD) resulted in a loss of ≤ 3% trials per study. Means and standard deviations for percent correct and reaction time were calculated as averaged performance over 120 trials. The correlation coefficient for RT hits and percent

**Table 3. a. Descriptive statistics and p-values for biodemographics, lifestyle and self-reported stress and anxiety by age group, by stress group for Study 1.** b. Descriptive statistics and p-values for biodemographics, lifestyle, self-reported stress and anxiety comparisons by age group, by stress group for Study 2A. c. Descriptive statistics and p-values for biodemographics, lifestyle, self-reported stress and anxiety comparisons by age group, by stress group for Study 2B. d. Means and standard errors of total cumulative stress scores for each study by age group by stress group.

| | | Young Adults | | | Older Adults | | |
|---|---|---|---|---|---|---|---|
| Study 1 (n = 40) | | Low Stress (n = 11) | High Stress (n = 10) | p | Low Stress (n = 10) | High Stress (n = 9) | p |
| | Age (years: mean (SD)) | 20.55 (2.54) | 22.00 (5.66) | 0.449[b] | 69.30 (8.31) | 68.89 (3.92) | 0.894[b] |
| | Gender (m:f) | 5:6 | 0:10 | 0.054[d] | 6:4 | 1:8 | 0.084[d] |
| | Education (years: mean (SD)) | 14.55 (1.63) | 15.50 (1.58) | 0.191[b] | 14.80 (2.04) | 14.56 (2.65) | 0.824[b] |
| | Cigarette consumption (typical daily n) | 0 | 3 | - | 0 | 0 | - |
| | Alcohol consumption (weekly units intake)[a] | 6.20 (2.54) | 1.63 (0.88) | 0.125[c] | 4.48 (2.57) | 2.56 (0.73) | 0.509[c] |
| | Exercise (hours per week)[a] | 2.91 (0.44) | 2.90 (0.44) | 0.988[b] | 2.20 (0.54) | 3.11 (0.49) | 0.237[b] |
| | Yoga (yes:no) | 2:9 | 3:7 | 0.903[d] | 0:10 | 1:8 | 0.212[d] |
| | Meditation (yes:no) | 0:11 | 4:6 | 0.076[d] | 2:8 | 0:9 | 0.503[d] |
| | Physical disability (yes:no) | 0 | 0 | - | 0 | 1 | - |
| | STAI—S[a] | 32.45 (2.49) | 34.30 (3.16) | 0.650[b] | 29.80 (2.25) | 29.56 (3.51) | 0.954[c] |
| | STAI—T[a] | 40.82 (3.19) | 39.80 (3.06) | 0.823[b] | 31.80 (3.14) | 32.89 (3.12) | 0.814[b] |
| | Perceived Stress Scale[a] | 13.00 (1.33) | 16.90 (1.69) | 0.078[b] | 9.00 (1.81) | 12.22 (2.37) | 0.298[b] |
| | | Young Adults | | | Older Adults | | |
| Study 2A (n = 58) | | Low Stress (n = 16) | High Stress (n = 15) | p | Low Stress (n = 14) | High Stress (n = 13) | p |
| | Age (years: mean (SD)) | 27.44 (4.07) | 29.67 (3.62) | 0.119[b] | 62.86 (2.88) | 65.46 (4.59) | 0.096[c] |
| | Gender (m:f) | 9:7 | 9:6 | 0.833[d] | 7:7 | 2:11 | 0.134[e] |
| | Education (years: mean (SD)) | 17.25 (2.05) | 16.00 (2.07) | 0.102[b] | 15.93 (2.70) | 16.46 (2.03) | 0.570[b] |
| | Cigarette consumption (typical daily n)[a] | 0.00 (0.00) | 1.67 (1.11) | 0.173[c] | 2.64 (1.37) | 1.23 (1.11) | 0.464[b] |
| | Alcohol consumption (weekly units intake)[a] | 1.45 (0.48) | 1.95 (0.92) | 0.622[b] | 4.63 (1.86) | 1.25 (0.62) | 0.099[c] |
| | Exercise (hours per week)[a] | 3.56 (0.39) | 3.67 (0.29) | 0.835[b] | 3.79 (0.34) | 3.62 (0.40) | 0.756[b] |
| | Yoga (yes:no) | 2:14 | 1:14 | 1[e] | 0:14 | 0:13 | - |
| | Meditation (yes:no) | 0:16 | 0:15 | - | 1:13 | 0:13 | - |
| | Physical disability (yes:no) | 0:16 | 0:15 | - | 1:13 | 1:12 | 1[e] |
| | STAI—S[a] | 33.75 (2.20) | 35.87 (2.89) | 0.567[b] | 27.00 (1.82) | 34.85 (3.82) | 0.084[c] |
| | STAI—T[a] | 41.13 (2.34) | 39.20 (3.24) | 0.635[b] | 32.64 (1.82) | 43.54 (3.68) | 0.025[c] |
| | Perceived Stress Scale (PSS-10)[a] | 14.31 (1.31) | 14.60 (2.02) | 0.918[b] | 11.36 (1.78) | 17.00 (2.21) | 0.064[b] |
| | | Young Adults | | | Older Adults | | |
| Study 2B (n = 58) | | Low Stress (n = 15) | High Stress (n = 14) | p | Low Stress (n = 15) | High Stress (n = 14) | p |
| | Age (years: mean (SD)) | 27.33 (5.45) | 28.50 (4.40) | 0.533[b] | 63.53 (3.85) | 65.93 (5.81) | 0.199[b] |
| | Gender (m:f) | 5:10 | 2:12 | 0.445[e] | 6:9 | 6:8 | 0.876[d] |
| | Education (years: mean (SD)) | 16.13 (2.26) | 16.00 (1.96) | 0.867[b] | 15.13 (2.03) | 15.64 (2.76) | 0.574[b] |
| | Cigarette consumption (typical daily n)[a] | 1.40 (1.03) | 0.07 (0.07) | 0.227[c] | 1.00 (0.97) | 0.00 (0) | 0.334[c] |
| | Alcohol consumption (weekly units intake)[a] | 1.28 (0.99) | 1.82 (0.76) | 0.674[b] | 4.75 (1.56) | 10.38 (2.51) | 0.077[c] |
| | Exercise (hours per week)[a] | 2.27 (0.39) | 2.50 (0.32) | 0.656[b] | 2.20 (0.37) | 3.64 (0.42) | 0.012[c] |
| | Yoga (yes:no) | 1:14 | 1:13 | 1[e] | 0:15 | 1:13 | 0.972[e] |
| | Meditation (yes:no) | 3:12 | 2:12 | 1[e] | 0:15 | 2:12 | 0.433[e] |
| | Physical disability (yes:no) | 0:15 | 1:13 | 0.972[e] | 1:14 | 2:12 | 0.950[e] |
| | STAI—S[a] | 35.80 (3.77) | 38.64 (2.35) | 0.532[b] | 25.40 (1.06) | 26.21 (1.68) | 0.692[b] |
| | STAI—T[a] | 44.53 (2.90) | 45.00 (2.91) | 0.923[b] | 30.93 (1.71) | 33.14 (2.36) | 0.466[b] |
| | Perceived Stress Scale (PSS-10)[a] | 15.20 (1.89) | 17.29 (1.65) | 0.420[b] | 9.33 (1.16) | 8.86 (1.43) | 0.805[b] |

| | Young Adults | | Older Adults | |
|---|---|---|---|---|
| | Low Stress | High Stress | Low Stress | High Stress |
| | mean (SE) | mean (SE) | mean (SE) | mean (SE) |
| Study 1 (n = 40) | 460.36 (44.02) | 691.70 (31.88) | 760.34 (40.79) | 993.11 (20.09) |
| Study 2A (n = 58) | 391.56 (31.95) | 703.53 (25.67) | 602.36 (38.49) | 846.46 (20.91) |
| Study 2B (n = 58) | 358.80 (19.49) | 706.14 (33.65) | 643.33 (32.78) | 860.86 (17.69) |

[a] Mean (SE). Standard error obtained via BCa Bootstrap with 1000 samples.

[b] p-value (two-tailed) calculated using an independent samples t-test.

[c] p-value (two-tailed) calculated using Welch's t-test.

[d] p-value (two-tailed) calculated using a chi-square test for independence.

[e] p-value (two-tailed) calculated using a chi-square test for independence (with Yates' Correction).

Young adults completed the Life Events Scale for Students (LESS). Score range: 0–1849.

Older adults completed the Social Readjustment Rating Scale (SRRS). Score range: 0–1466.

correct hits was consistently negative and not statistically significant (p's ≥ 0.179) for all studies indicating that a speed-accuracy trade-off would be unlikely to bias the planned statistical analyses.

We proceeded with the median split because age group and stress group were not significantly correlated (p > .9). Given that the median split between high and low stress groups by study led to an overlap in classification of high and low stress between the studies, a single median split was also derived by grouping all studies' observations. For the LESS the single median split was 577 and for the SRRS, 786. For ease of reference, a table with all the high vs. low median split values is provided in Supporting Information (S4 Table). Using the single median split, we conducted the same statistical procedures as for our planned analyses, described in 'Design and Statistical Analysis', which used the median split derived by study.

Table 4. Age group percent correct and RT mean differences, standard errors, credible intervals and Bayes factors for all studies.

| | | Young vs. Older Adults | | | | |
|---|---|---|---|---|---|---|
| Accuracy (% correct) | | Prior | Likelihood | Posterior | | |
| | N: Incremental Increase | mean Difference (SE) | mean Difference (SE) | mean Difference (SE) | 95% credible interval[a] | BF |
| Marshall et al. (N = 60) | 60 | 5.00[a] | 4.83 (1.64) | | | 15.99[†] |
| Study 1 (N = 40) | 100 | 4.83 (1.64) | 4.35 (2.42) | 4.68 (1.36) | 2.02, 7.34 | 1.59 |
| Study 2A (N = 58) | 158 | 4.68 (1.36) | 1.29 (2.12) | 3.70 (1.14) | 1.46, 5.94 | 0.58 |
| Study 2B (N = 58) | 216 | 3.70 (1.14) | -1.98 (2.06) | 2.37 (1.00) | 0.41, 4.33 | 0.78 |
| Reaction Time (ms) | | mean Difference (SE) | mean Difference (SE) | mean Difference (SE) | 95% credible interval[a] | BF |
| Marshall et al. (N = 46) | 46 | 50.00[b] | -441.28 (109.07) | | | 3.77[†] |
| Study 1 (N = 40) | 86 | -241.43 (59.76) | -358.56, -124.30 | -241.43 (59.76) | -358.56, -124.30 | 2.64 |
| Study 2A (N = 58) | 144 | -165.46 (42.35) | -248.47, -82.44 | -165.46 (42.35) | -248.47, -82.44 | 1.22 |
| Study 2B (N = 58) | 202 | -124.96 (31.40) | -186.50, -63.42 | -124.96 (31.40) | -186.50, -63.42 | 1.33 |

[a] In the first iteration, an estimated maximum performance difference of 10% was assumed. Half of this value (5%) was used as a vague prior. This prior was used to calculate the BF for Marshall et al. (2015)'s [38] result.

[b] In the first iteration, an estimated maximum performance difference of 100 ms was assumed. Half of this value (50 ms) was used as a vague prior. This prior was used to calculate the BF for Marshall et al. (2015)'s [38] result.

[†] evidence favours H1 (BF > 3).

[‡] evidence favours H0 (BF < ⅓).

**Table 5. Cumulative stress percent correct and RT mean differences, standard errors, credible intervals and Bayes factors for all studies.**

| | | Low Stress vs. High Stress | | | | |
|---|---|---|---|---|---|---|
| *Accuracy (% correct)* | | *Prior* | *Likelihood* | *Posterior* | | |
| | *N: Incremental Increase* | *mean Difference (SE)* | *mean Difference (SE)* | *mean Difference (SE)* | *95% credible interval*[a] | *BF* |
| Marshall et al. (N = 60) | 60 | 5.00[a] | 3.50 (1.62) | | | 2.54 |
| Study 1 (N = 40) | 100 | 3.50 (1.62) | 5.77 (2.35) | 4.23 (1.33) | 1.62, 6.85 | 2.36 |
| Study 2A (N = 58) | 158 | 4.23 (1.33) | 3.79 (2.2) | 4.11 (1.14) | 1.88, 6.35 | 1.55 |
| Study 2B (N = 58) | 216 | 4.11 (1.14) | -1.18 (2.07) | 2.88 (1.00) | 0.93, 4.84 | 0.61 |
| *Reaction Time (ms)* | | *mean Difference (SE)* | *mean Difference (SE)* | *mean Difference (SE)* | *95% credible interval*[a] | *BF* |
| Marshall et al. (N = 46) | 46 | 50.00[b] | 62.33 (158.09) | | | 0.96 |
| Study 1 (N = 40) | 86 | 62.33 (158.09) | -146.53 (72.13) | -110.54 (65.62) | -239.16, 18.08 | 1.27 |
| Study 2A (N = 58) | 144 | -110.54 (65.62) | 64.12 (59.29) | -14.38 (43.99) | -100.61, 71.85 | 1.00 |
| Study 2B (N = 58) | 202 | -14.38 (43.99) | 2.63 (48.69) | -6.73 (32.64) | -70.72, 57.25 | 0.81 |

[a] In the first iteration, an estimated maximum performance difference of 10% was assumed. Half of this value (5%) was used as a vague prior. This prior was used to calculate the BF for Marshall et al. (2015)'s [38] result.

[b] In the first iteration, an estimated maximum performance difference of 100 ms was assumed. Half of this value (50 ms) was used as a vague prior. This prior was used to calculate the BF for Marshall et al. (2015)'s [38] result.

[†] evidence favours H1 (BF > 3).

[‡] evidence favours H0 (BF < ⅓).

We found the outcomes for both sets of analyses to be comparable, therefore we report the results for our planned analysis here and provide the results for the identical analysis using the single median split in Supporting Information (S5 Table).

Table 4 presents the meta-analysis showing the effect of age on accuracy and reaction time performance. We also provide an appendix with a set of tables with all data entered into each analysis in the (S5 Appendix). Table 4 shows that as each study's data was added and the sample size increased, the effect size decreased. The final credible interval ranged between .4% and 4.3%. Had there been robust evidence to show that young adults outperformed older adults, the effect size and corresponding BFs would have increased with each newly added dataset. Inspecting the BFs in the table, Marshall et al.'s reported effect size was supported by strong evidence (BF = 15.99) for a difference in accuracy between young and older adults with young adults out-performing older adults. However, with the additional data, BFs decreased, providing anecdotal evidence only (BFs ≤ 1.59). A likewise outcome was evident for reaction time: Marshall et al.'s reported result of a difference in response latencies between older and young adults was supported by substantial evidence (BF = 3.77) indicating that young adults were significantly faster than older adults. However, as with accuracy, each dataset added incrementally lead to a smaller rather than larger effect size. Our final credible interval ranged from -186.5 ms to -63.42 ms with a corresponding BF supported by anecdotal evidence (BF = 1.33).

Table 5 presents the results for the overall comparison of low vs. high cumulative life stress groups for accuracy and reaction time. Marshall et al.'s reported result provided anecdotal evidence for accuracy (BF = 2.54) and reaction time (BF = 0.96) indicating that it is unclear whether or not high levels of cumulative life stress affect performance outcomes. The table shows that by adding new data and additional power this outcome remained consistently within the anecdotal range (BFs ≤ 1.27). For accuracy, the final credible interval for accuracy ranged between 0.9% and 4.8%. For reaction time, the range was -70.7 ms to 57.3 ms.

Table 6 presents the outcome for the interaction effect. Marshall et al. critically found a statistically significant interaction effect for accuracy, which was supported by strong evidence (BF = 50.86) as shown in the table. However, with each iteration, the effect size decreased and

**Procedure**

**Study 1, 2A and 2B**

- Welcome
- Information about the tasks, procedures, what to expect
- Exclusion Criteria checked (in addition to pre-attendance screening)
- Informed Consent

**Self-report measures**
- Biodemographic Information questions
- Health and lifestyle questions
- Life Events Questionnaire:
  - Life Events Scale for Students (LESS): 18-35 yrs
  - Social Readjustment Rating Scale (SRRS): 60 – 85 yrs
- Pittsburgh Sleep Quality Index
- Perceived Stress Scale (PSS-10)
- Brief Resilience Scale (BRS)
- STAI-S
- STAI-T

**Cognitive Task: n-back**

Study 1

- 1-BACK Practice:
  - Block 1 [20 trials]
- 2-BACK Practice:
  - Block 1 [20 trials]

- Free-Recall Task
- 2-BACK Experimental Trials:
  - Block 1 [40 trials]
  - Block 2 [40 trials]
  - Block 3 [40 trials]

Study 2A

- 1-BACK Practice:
  - Block 1 [20 trials]
- 2-BACK Practice:
  - Block 1 [20 trials]
- 2-BACK Experimental Trials:
  - Block 1 [40 trials]
  - Block 2 [40 trials]
  - Block 3 [40 trials]

Study 2B

- 1-BACK Practice:
  - Block 1 [20 trials]
- 1-BACK Experimental Trials:
  - Block 1 [40 trials]
  - Block 2 [40 trials]
  - Block 3 [40 trials]

- 2-BACK Practice:
  - Block 1 [20 trials]
- 2-BACK Experimental Trials:
  - Block 1 [40 trials]
  - Block 2 [40 trials]
  - Block 3 [40 trials]

**Fig 1. Procedure.** A graphic representation of each study's procedure and tasks administered.

**Table 6. Percent correct and RT mean differences, standard errors, credible intervals and Bayes factors for young low and high stress groups by older low and high stress groups interaction effect for all studies.**

| | | Age by Stress Group Interaction | | | | |
| --- | --- | --- | --- | --- | --- | --- |
| | | *Prior* | *Likelihood* | *Posterior* | | |
| *Accuracy (% correct)* | *N: Incremental Increase* | *mean Difference (SE)* | *mean Difference (SE)* | *mean Difference (SE)* | *95% credible interval[a]* | *BF* |
| **Marshall et al. (N = 60)** | **60** | 2.50[a] | -12.55 (2.85) | | | 50.86[†] |
| **Study 1 (N = 40)** | **100** | -12.55 (2.85) | 1.53 (4.65) | -8.70 (2.43) | -13.47, -3.94 | 0.62 |
| **Study 2A (N = 58)** | **158** | -8.70 (2.43) | -1.54 (4.23) | -6.92 (2.11) | -11.05, -2.79 | 0.92 |
| **Study 2B (N = 58)** | **216** | -6.92 (2.11) | 6.39 (4.23) | -4.28 (1.89) | -7.98, -0.58 | 1.06 |
| *Reaction Time (ms)* | | *mean Difference (SE)* | *mean Difference (SE)* | *mean Difference (SE)* | *95% credible interval[a]* | *BF* |
| **Marshall et al. (N = 46)** | **46** | 25.50[b] | -47.75 (254.72) | | | 1.00 |
| **Study 1 (N = 40)** | **86** | -47.75 (254.72) | 104.80 (138.22) | 70.10 (121.49) | -168.01, 308.22 | 0.99 |
| **Study 2A (N = 58)** | **144** | 70.10 (121.49) | 93.36 (123.58) | 81.53 (86.64) | -88.27, 251.33 | 0.74 |
| **Study 2B (N = 58)** | **202** | 81.53 (86.64) | -19.04 (96.41) | 36.60 (64.44) | -89.70, 162.90 | 0.62 |

[a] In the first iteration, an estimated maximum performance difference of 5% was assumed. Half of this value (2.5%) was used as a vague prior. This prior was used to calculate the BF for Marshall et al. (2015)'s [38] result.

[b] In the first iteration, an estimated maximum performance difference of 50 ms was assumed. Half of this value (25 ms) was used as a vague prior. This prior was used to calculate the BF for Marshall et al. (2015)'s [38] result.

[†] evidence favours H1 (BF > 3).

[‡] evidence favours H0 (BF < ⅓).

the BF provided anecdotal evidence only (BFs ≤ 1.06). Our final credible interval for accuracy was .9% to 4.8% with a corresponding BF of 1.06. Marshall et al. did not find a statistically significant interaction for reaction time, which corresponded with our BF outcome (BFs ≤ 1.00). Our final credible interval for RT was -89.7 ms to 162.9 ms with a corresponding BF of 0.62. Hence, the interaction effect between age and stress is not robust.

Table 7A and 7B provide additional information for low vs. high stress groups within age group. In each table, regardless of the initial BF and effect size obtained for Marshall et al., the respective final outcomes indicate anecdotal evidence.

Sensitivity analyses revealed that all of the above reported BFs but one were consistent across a range of scale factors, which indicates that the obtained BF values are robust. The exception was the comparison of low vs. high stress groups (Step 1) where the BF was 2.36. Here the BF showed substantial evidence with the Student's t (BF = 3.08) and Cauchy (BF = 3.28) distributions but subsequently dropped back into the anecdotal range (BFs ≤ 1.55) as more data were added. On this basis, the BF for the overall comparison of low and high stress groups is still fairly robust.

Post hoc, we assessed whether WM performance varied as a consequence of study mode (in-person vs. online) given that Marshall et al.'s [38] study and Study 1 were in-person studies whereas Studies 2A and 2B were conducted online. To evaluate whether accuracy (n-back percent correct, d-prime) and/or n-back RT varied as a function of study mode (in-person: Marshall, Study 1 vs. online: Study 2A, Study 2B) we conducted an independent samples t-test for each dependent variable. The result showed that accuracy scores were comparable (p's ≥ .099); however for RTs the result was statistically significant: t(118) 2.940, p = .004. Corresponding BFs for percent correct and d-prime indicated evidence for the null (BF = 0.21) and the insensitive range bordering on the null (BF = 0.34), respectively. For RTs there was strong evidence supporting an effect (BF = 17.53). Bayes Factors were calculated in JASP [92]. These results indicate that participants who took part in-person were slower (M 853.48, SE 44.99) on average than online participants (M 709.84, SE 19.51). Reasons may include that the online

**Table 7. a.YA percent correct and RT mean differences, standard errors, credible intervals and Bayes factors by stress group within age group for all studies.** b. OA percent correct and RT mean differences, standard errors, credible intervals and Bayes Factors by stress group within age group for all studies.

| | | YA: Low vs. High Stress | | | | |
| --- | --- | --- | --- | --- | --- | --- |
| | | *Prior* | *Likelihood* | *Posterior* | | |
| *Accuracy (% correct)* | *N: Incremental Increase* | *mean Difference (SE)* | *mean Difference (SE)* | *mean Difference (SE)* | *95% credible Interval[a]* | *BF* |
| **Marshall et al. (N = 60)** | **60** | 5.00[a] | -2.39 (1.88) | | | 0.71 |
| **Study 1 (N = 40)** | **100** | -2.39 (1.88) | 6.13 (3.12) | -0.13 (1.61) | -3.28, 3.03 | 1.20 |
| **Study 2A (N = 58)** | **158** | -0.13 (1.61) | 3.16 (3.06) | 0.59 (1.42) | -2.21, 3.38 | 1.00 |
| **Study 2B (N = 58)** | **216** | 0.59 (1.42) | 1.79 (3.60) | 0.75 (1.32) | -1.85, 3.34 | 0.93 |
| *Reaction Time (ms)* | | *mean Difference (SE)* | *mean Difference (SE)* | *mean Difference (SE)* | *95% credible Interval[a]* | *BF* |
| **Marshall et al. (N = 46)** | **46** | 12.50[b] | -87.32 (78.59) | | | 1.00 |
| **Study 1 (N = 40)** | **86** | -87.32 (78.59) | -111.89 (109.76) | -95.65 (63.90) | -220.89, 29.60 | 1.00 |
| **Study 2A (N = 58)** | **144** | -95.65 (63.90) | 121.94 (82.12) | -13.59 (50.43) | -112.44, 85.26 | 1.02 |
| **Study 2B (N = 58)** | **202** | -13.59 (50.43) | -25.33 (65.47) | -17.96 (39.95) | -96.27, 60.35 | 0.86 |
| | | OA: Low vs. High Stress | | | | |
| | | *Prior* | *Likelihood* | *Posterior* | | |
| *Accuracy (% correct)* | *N: Incremental Increase* | *mean Difference (SE)* | *mean Difference (SE)* | *mean Difference (SE)* | *95% credible Interval[a]* | *BF* |
| **Marshall et al. (N = 60)** | **60** | 12.50[a] | 9.39 (2.01) | | | >100[†] |
| **Study 1 (N = 40)** | **100** | 9.39 (2.01) | 5.39 (3.26) | 8.29 (1.71) | 4.93, 11.64 | 1.43 |
| **Study 2A (N = 58)** | **158** | 8.29 (1.71) | 4.53 (2.83) | 7.28 (1.47) | 4.4, 10.15 | 1.23 |
| **Study 2B (N = 58)** | **216** | 7.28 (1.47) | -4.15 (2.14) | 3.62 (1.21) | 1.25, 5.99 | 1.94 |
| *Reaction Time (ms)* | *N: Incremental Increase* | *mean Difference (SE)* | *mean Difference (SE)* | *mean Difference (SE)* | *95% credible Interval[a]* | *BF* |
| **Marshall et al. (N = 46)** | **46** | 37.50[b] | 102.87 (209.13) | | | 0.99 |
| **Study 1 (N = 40)** | **86** | 102.87 (209.13) | -185.66 (75.62) | -152.3 (71.11) | -291.68, -12.92 | 2.15 |
| **Study 2A (N = 58)** | **144** | -152.3 (71.11) | -2.73 (93.16) | -97.23 (56.53) | -208.03, 13.56 | 1.00 |
| **Study 2B (N = 58)** | **202** | -97.23 (56.53) | 30.59 (69.27) | -46.14 (43.79) | -131.98, 39.7 | 0.98 |

[a] In the first iteration, an estimated maximum performance difference of 10% was assumed. Half of this value (5%) was used as a vague prior. This prior was used to calculate the BF for Marshall et al. (2015)'s [38] result.

[b] In the first iteration, an estimated maximum performance difference of 25 ms was assumed. Half of this value (12.5 ms) was used as a vague prior. This prior was used to calculate the BF for Marshall et al. (2015)'s [38] result.

[†] evidence favours H1 (BF > 3).

[‡] evidence favours H0 (BF < ⅓).

participants were more computer literate, making them faster or they were less stressed. As such, this result suggests that we can only validly compare Study 1's RT results with Marshall et al.'s work. However, all our statistical results and consequent conclusions are the same whether we include or exclude the online (Study 2A, 2B) results. Tables 4–7 illustrate this.

## Discussion

The aim of the present study was to assess the strength of the evidence supporting the hypothesis that higher cognitive function, as measured by working memory, is negatively affected by cumulative life stress, ageing and/or the interaction of the two. Marshall and colleagues [38] conducted a study with 60 participants which showed a robust age by cumulative stress effect where only older participants with a higher cumulative stress score showed impaired performance on a 2-back (and Sternberg, [93]) task. We were unable to replicate this result using an iterative Bayesian meta-analysis, comprising three replication studies ($N_{total} = 156$) with the same 2-back task and cumulative stress measures. Indeed for all 3 effects investigated, namely, ageing, cumulative stress and their interaction, our results fell within the anecdotal range (⅓<BF<3).

Our analysis of the age by cumulative stress interaction revealed a finding within the anec-dotal range for accuracy and RT. In particular, where Marshall et al.'s study shows a clear detri-ment in accuracy of older high stress participants compared to older low stress participants and no detectable effect of stress in the young low vs. high stress groups, we found no such interaction. We found that for both accuracy and RT, older adults showed inconclusive evi-dence of an effect of cumulative stress. Likewise for young adults. Thus, our study does not support the hypothesis that high stress older adults show a particular impairment due to the cumulative impact of stress on higher cognitive function, as measured in working memory.

Our inconclusive result for cumulative life stress is incongruent with Marshall et al.'s study which found no evidence of an effect; from a Bayesian perspective our results indicate that additional data is needed to provide evidence to confirm either the null or an effect of stress. One possible reason for these results is that ageing is a powerful mediator and may have masked the effect of cumulative stress on cognition. For example, in a meta-analysis investigat-ing the impact of processing speed on cognition, Verhaeghen [94] found that ageing mediated performance on a range of cognitive functions such as working memory, executive control and task shifting, explaining $\leq$ 58% of age-related variance. In light of such evidence, one may speculate that YA performance may better represent the impact of cumulative stress on cogni-tion without the added variance contributed by ageing effects on performance. Indeed, studies of YAs that have investigated the effects of recent life events stress (4 to 6 months) found that high stress YAs showed blunted autonomic responsivity [95] and had poorer academic perfor-mance, psychological and physical health [96] than low stress YA. Low stress YAs were also better able to avoid risks when making decisions [97]. Given that we evaluated a YA sample and their results also fell within the insensitive range suggests that how we measured the impact of cumulative life stress on working memory may not have been sufficiently reliable relative to our sample size.

Our overall comparison by age group was inconclusive for both accuracy and RT. The find-ing for RT in particular was unexpected given that processing speed in the context of higher cognitive function draws on a wide range of neural networks, which, as mentioned in the introduction, deteriorate with advancing age [13,14,98–100]. Indeed, Verhaeghen [94] found processing speed explained 78% of the variance in cognitive functions. Our finding was also at odds with Marshall et al.'s results, which showed that older adults were slower and less accu-rate than young adults. Using Bayesian methods, our results suggest that more data is needed.

All 3 of our replication studies produced consistently small effect sizes which is incongruent with the much larger effects we expected based on Marshall et al.'s findings. There are a num-ber of factors that may explain our outcomes. Older adults' stress responsivity appears to be less efficient and consequently may have relatively limited impact on memory performance [101]. Individual differences, too, play a role regarding both cognitive ability [94] and basal cortisol levels [102]. Previous research found that working memory declines with age but interacts with sex and education [103] and that processing complexity is more vulnerable to ageing than the (passive) storage components of working memory [104]. Our hierarchical lin-ear regressions (unpublished data) conducted for each of our replication studies separately, with stress scores as a continuous variable, showed no impact of age group or education on performance, nor indeed other factors such as gender, exercise, perceived stress or alcohol consumption. Was there a speed-accuracy trade-off? Our analyses suggests not, however we note that, compared to our young adults, there was more variability in older adults' scores and they do tend to be more cautious when providing responses [105].

Our study's findings should be interpreted alongside some caveats. While a meta-analysis has advantages in providing better statistical power, we cannot rule out the impact of con-founding moderating/mediating factors. In particular, Study 1 and Marshall et al.'s study were

conducted in-person whilst Studies 2A and 2B were performed online. In addition, the Covid-19 pandemic occurred around the time of our 3 studies. Our iterative approach allows the effects of each study to be considered separately and we found no evidence to indicate that these factors had an impact on our conclusions. Our data comprised retrospectively collected life events information, which are only an indicator of the accumulated stress impact that people have experienced. That we failed to find conclusive evidence of an age by cumulative stress interaction effect is likely to be because our study was underpowered. When we designed our first study, we believed it to be well-powered as Marshall's study demonstrated a robust interaction effect with a much smaller sample (60 vs. 156). Moreover, we used an iterative Bayesian meta-analysis, which displays the pattern of the effect over studies in addition to providing a pooled effect size. Bayesian statistics is not subject to the stopping rule of frequentist methods and therefore the number of participants enrolled for a study can be incrementally increased until the data are sensitive enough to reveal sufficient evidence to confidently conclude an outcome in favour of either H1 or H0 [65] or neither. Throughout, our effect sizes became progressively smaller with each incremental increase in sample size, which was comparatively large. We note that our meta-analysis assumed that there is one effect size being measured, however given the consistency of our effect sizes with three independent samples, we believe this assumption is valid here. We analysed the data using a common median split and median split by sample, with very similar results, which provides additional assurance as to the sensitivity of our analysis. This is in addition to sensitivity analysis conducted to confirm the robustness of BFs reported. Thus, we are confident in our findings.

In conclusion, our Bayesian meta-analysis suggests inconclusive evidence for the effect of ageing, cumulative stress and their interaction on working memory, as measured with a life events questionnaire. The design of our study at the outset was well-powered given the previous research. Our results, however, indicate that this was not the case and we argue that using a life events questionnaire to evaluate the impact of cumulative life stress on working memory with relatively small sample sizes will not reliably capture any effect that might exist because there are too many extraneous factors such as individual differences, developmental factors or epigenetics [106,107].

## Supporting information

**S1 Table. Age median IQR values for participants for all 3 studies.**
(PDF)

**S2 Table. Analyses for percent correct and D-prime means, SEs and Univariate ANOVA results for all 3 studies.**
(PDF)

**S3 Table. Frequency table of the total cumulative stress score for each participant in each study.**
(XLSX)

**S4 Table. Reference table providing the median splits for all 3 studies.**
(PDF)

**S5 Table. Sensitivity analysis: A comparison using a single median split.**
(PDF)

**S1 Appendix. Participant details for all 3 studies.**
(PDF)

**S2 Appendix. Additional tasks administered during the study (description and results).**
(PDF)

**S3 Appendix. An excel workbook containing a series of tables showing power analysis and traditional meta-analysis outcomes.**
(XLSX)

**S4 Appendix. Study design and procedure for all 3 studies.**
(PDF)

**S5 Appendix. Data tables for all iterative analyses and Bayes factor calculations.**
(XLSX)

## Author Contributions

**Conceptualization:** Denise Wallace, Nicholas R. Cooper, Alejandra Sel, Riccardo Russo.

**Data curation:** Denise Wallace.

**Formal analysis:** Denise Wallace, Nicholas R. Cooper, Alejandra Sel, Riccardo Russo.

**Funding acquisition:** Denise Wallace.

**Investigation:** Riccardo Russo.

**Methodology:** Denise Wallace, Nicholas R. Cooper, Alejandra Sel, Riccardo Russo.

**Project administration:** Denise Wallace.

**Resources:** Denise Wallace.

**Supervision:** Nicholas R. Cooper, Alejandra Sel, Riccardo Russo.

**Validation:** Nicholas R. Cooper, Alejandra Sel, Riccardo Russo.

**Visualization:** Denise Wallace.

**Writing – original draft:** Denise Wallace.

**Writing – review & editing:** Denise Wallace, Nicholas R. Cooper, Alejandra Sel, Riccardo Russo.

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
