## [Decision Letter · Decision Letter 0]

1 Feb 2023

PONE-D-22-27705Do non-traumatic stressful life events and ageing negatively impact working memory performance and do they interact to further impair working memory performance?PLOS ONE

Dear Dr. Wallace,

Thank you for submitting your manuscript to PLOS ONE. After careful consideration, we feel that it has merit but does not fully meet PLOS ONE’s publication criteria as it currently stands. Therefore, we invite you to submit a revised version of the manuscript that addresses the points raised during the review process. Please consider that all the comments and indications raised by Reviewers are very important and need to be addressed in the revised version of the manuscript. Furthermore, please pay attention to submission guidelines. The abstract, for example, should not exceed 300 words – please check and modify it accordingly. Be aware that not respecting journal guidelines may cause delay in the review process.

We look forward to receiving your revised manuscript.

Kind regards,

Irene Ronga, Ph.D.

Academic Editor

PLOS ONE

and https://journals.plos.org/plosone/s/file?id=ba62/PLOSOne_formatting_sample_title_authors_affiliations.pdf.

2. Please amend the manuscript submission data (via Edit Submission) to include author Nicholas R. Cooper, Alejandra Sel and Riccardo Russo.

Reviewers' comments:

Reviewer's Responses to Questions

**Comments to the Author**

1. Is the manuscript technically sound, and do the data support the conclusions?

Reviewer #1: Yes

Reviewer #2: Partly

2. Has the statistical analysis been performed appropriately and rigorously? 

Reviewer #1: Yes

Reviewer #2: Yes

3. Have the authors made all data underlying the findings in their manuscript fully available?

Reviewer #1: Yes

Reviewer #2: No

4. Is the manuscript presented in an intelligible fashion and written in standard English?

Reviewer #1: Yes

Reviewer #2: Yes

5. Review Comments to the Author

Reviewer #1: The study is interesting and well-described.

However, here are some considerations:

There are some typos (e.g. line 409, insert complete reference or reference in the bibliography - Sternberg)

Concerning sample recruitment, there is no mention of the Covid- 19 period.

As the samples were recruited in different periods, could the historical context (e.g. isolation, stress, fear, uncertainty etc., due to the pandemic) have influenced the research results?

Furthermore, could these aspects have influenced the difference in the results obtained compared to the research by Marshall et al.,2015?

Since one study was conducted in presence while the other two were conducted online, could the mode influence the results?

Reviewer #2: The authors' replication attempt of Marshall et al. (2015) is interesting both because of the importance of replication effort and because the robustness of their chosen statistical approach is supported by robustness checks as implemented by frequentists meta-analyses.

I am, however, not completely convinced by their choice of splitting the participants based on the median scores on the tests. For instance, when analyzing the SRRS a previous study (Hobson, C.J., Delunas, L. 2001) finds a mean score of 278 based on a sample of 3399 returned surveys (out of 5000 sent to a representative sample). While the scale is not straightforward in its interpretation, a score of over 300 is interpreted as evidence of major stress (Fontecilla et al, 2012). However, the authors report higher scores - are all of their older participants severely stressed? I understand that using a cutoff of 300 might have meant having a very unbalanced design, but this point should be discussed, at minimum by providing a breakdown of severely stressed participants by subgroup (young adults vs older adults) and the possible implications on the results.

Other notes follow:

Lines 416-423: Indeed, the lack of interaction (whether computed by means of bayesian or frequentist meta-analysis) seems to confirm that ageing and stress may independently impair WM performance. However, I am concerned about the appropriateness of describing what amounts to an interaction effect (lines 426-427) after having shown that there is no interaction effect. I would limit the discussion of the effect on stress on young adults to the previous paragraph (around lines 412-413), expanding by referencing works that investigate the effects of stress on young adults cognition.

Alternatively, the same topic (effect of stress on YA) could be discussed around lines 444-446, when the topic is brought up again.

Furthermore, line 415: ... the evidence breached the insensitive range. Did the authors mean 'remained in the insensitivity range'?

Finally, I believe the tables should be checked again for errors. The authors should either report all p values or mark the significant differences in the tables using formatting, though the first option would greatly be preferred. I find that reporting only the lowest p-value for each row creates a bit of confusion, and adds little information (knowing that p values are greater than .3 does not give more insight that knowing that the p values are greater than .05)

In the same way, it is not clear why Mann-Whitney U tests were mentioned in the tables, but their values were not shown. It is possible that the authors chose to perform M-W U tests to check the results of a t-test, given possible deviations from t-test assumptions. However, deviations from normality do not imply that M-W U should be used (Fay, M.P. and Proschan, M.A. (2010)), and performing multiple tests does not lower the probability of type I errors. On the other hand, the authors did not specify whether Welch t-tests were used (this is preferred over performing a test for equality of variances followed by a t-test - Fay And Proschnan, 2010). If the choice of M-W U tests was on theoretical grounds, they should be reported instead of t-tests, not as robustness checks. Furthermore, it is unclear whether M-W U tests are significant whenever t tests are (the phrasing "similar outcomes" would suggest they are, but the p values are not reported)

Some typos or mispellings found in tables include the last cell of Table 1a, which has a typo (0.0.78). I would also rephrase 'Chi-Square test for independence (low vs high stress) were performed by age group (p-value represents Yates' Continuity Correction)' as 'Chi-Square tests for independence (with Yates' Correction) were performed for each age group'. In tables 2 to 4, Credibility Interval should maybe be spelled as 'credible interval'.

Furthermore, In tables 2, 3 and 4 Bayes Factors are reported. However, Table 4 reports the common thresholds used to support H1 and H0 (3 and 1/3), while Table and 2 3 do not, and do not mark BF lower than 1/3 or greater than 3.

Finally, If reporting p values as suggested before, asterisks could be removed. However, a doubt remains, as it is not clear what are the significance tests used from table 2 onwards and marked with asterisks.

6. PLOS authors have the option to publish the peer review history of their article (what does this mean?). If published, this will include your full peer review and any attached files.

Reviewer #1: No

Reviewer #2: No

---

## [Author Response · Author response to Decision Letter 0]

14 Apr 2023

and https://journals.plos.org/plosone/s/file?id=ba62/PLOSOne_formatting_sample_title_authors_affiliations.pdf.

2. Please amend the manuscript submission data (via Edit Submission) to include author Nicholas R. Cooper, Alejandra Sel and Riccardo Russo.

We have updated the manuscript to meet with the formatting requirements as indicated in points 1 - 3.

Reviewer comments with Authors’ replies:

Reviewer #1:

1. There are some typos (e.g. line 409, insert complete reference or reference in the bibliography - Sternberg)

Author reply: Thank you for noticing the missing reference. We have added a reference for Sternberg in the text (line 472) and in the bibliography (line 781). We have also attempted to address all typos. The changes in the manuscript to incorporate the reviewer’s suggestions read as follows:

“…where only older participants with a higher cumulative stress score showed impaired performance on a 2-back (and Sternberg,[93]) task. We conducted an iterative Bayesian meta-analysis,…”.

2. Concerning sample recruitment, there is no mention of the Covid- 19 period.

As the samples were recruited in different periods, could the historical context (e.g. isolation, stress, fear, uncertainty etc., due to the pandemic) have influenced the research results?

Furthermore, could these aspects have influenced the difference in the results obtained compared to the research by Marshall et al.,2015?

Since one study was conducted in presence while the other two were conducted online, could the mode influence the results?

Author reply: Thank you for this insightful comment. We have now added a paragraph in the Introduction (lines 129-138) and the Discussion (lines 526-532) which addresses the Covid-19 aspect. Regarding study mode, we have addressed this in the Results section (lines 449-465) and in the Discussion section (lines 526-532).

The changes in the manuscript to incorporate the reviewer’s suggestions read as follows (in line-order):

[lines 129-138] “We note that our research was conducted around the period of the Covid-19 pandemic, which was likely to have been particularly stressful given the level of disruption and uncertainty at the time. From this perspective, Marshall and colleagues’ (38) study was conducted well before the pandemic while our first study (Study 1) was conducted from September 2019 to March 2020, just before the first ever UK Covid-19 lockdown (64). Our subsequent 2 studies, which were run in parallel (Study 2A and Study 2B), followed in April 2021 just after the easing of full lockdown measures. One would therefore expect stress to be greater in our three studies relative to Marshall’s study and, consequently, finding an interaction between ageing and stress may arguably be more likely. However, given that our individual study results did not find evidence of any interaction we believe our results are robust.”

 [lines 449-465] “Post hoc, we assessed whether WM performance varied as a consequence of study mode (in-person vs. online) given that Marshall et al.’s (38) study and Study 1 were in-person studies whereas Studies 2A and 2B were conducted online. To evaluate whether accuracy (n-back percent correct, d-prime) and/or n-back RT varied as a function of study mode (in-person: Marshall, Study 1 vs. online: Study 2A, Study 2B) we conducted an independent samples t-test for each dependent variable. The result showed that accuracy scores were comparable (p’s ≥ .099); however for RTs the result was statistically significant: t(118) 2.940, p = .004. Corresponding BFs for percent correct and d-prime indicated evidence for the null (BF = 0.21) and the insensitive range bordering on the null (BF = 0.34), respectively. For RTs there was strong evidence supporting an effect (BF = 17.53). Bayes Factors were calculated in JASP (92). These results indicate that participants who took part in-person were slower (M 853.48, SE 44.99) on average than online participants (M 709.84, SE 19.51). Reasons may include that the online participants were more computer literate, making them faster or they were less stressed. As such, this result suggests that we can only validly compare Study 1’s RT results with Marshall et al.’s work. However, all our statistical results and consequent conclusions are the same whether we include or exclude the online (Study 2A, 2B) results. Tables 4 – 7 illustrate this.”

[lines 526-532] “Our study’s findings should be interpreted alongside some caveats. While a meta-analysis has advantages in providing better statistical power, we cannot rule out the impact of confounding moderating/mediating factors. In particular, Study 1 and Marshall et al.’s study were conducted in-person whilst Studies 2A and 2B were performed online. In addition, the Covid-19 pandemic occurred around the time of our 3 studies. Our iterative approach allows the effects of each study to be considered separately and we found no evidence to indicate that these factors had an impact on our conclusions. Our data….”

Reviewer #2: 

1. The authors' replication attempt of Marshall et al. (2015) is interesting both because of the importance of replication effort and because the robustness of their chosen statistical approach is supported by robustness checks as implemented by frequentists meta-analyses.

I am, however, not completely convinced by their choice of splitting the participants based on the median scores on the tests. For instance, when analyzing the SRRS a previous study (Hobson, C.J., Delunas, L. 2001) finds a mean score of 278 based on a sample of 3399 returned surveys (out of 5000 sent to a representative sample). While the scale is not straightforward in its interpretation, a score of over 300 is interpreted as evidence of major stress (Fontecilla et al, 2012). However, the authors report higher scores - are all of their older participants severely stressed? I understand that using a cutoff of 300 might have meant having a very unbalanced design, but this point should be discussed, at minimum by providing a breakdown of severely stressed participants by subgroup (young adults vs older adults) and the possible implications on the results.

Author reply: Thank you for raising this important issue. The score of 300 is an indicator of major stress based on the last 12 months. Rahe’s (1975) “Life changes and near-future illness reports” article states that a score of 300 or more = 80% chance of illness in the near future. In the present study, the total score represents a lifetime score. Consequently, the vast majority of participants exceed 300 and this is not an indicator of major stress. We think that it is important to demonstrate that our study is balanced. To address this we have added a supplemental frequency table called “S5” (lines 291-293) which provides all our participants’ cumulative stress scores by study. The supplemental table (S5) includes the frequency of each score and the cumulative percentage. Moreover, we have added to the manuscript Table 3d (lines 379-381), which gives the means and standard errors for each study, split by age group and by stress group to demonstrate that no strong biases exist in any of the sub-groups. 

The changes in the manuscript read as follows:

[lines 291-293] “…our hypothesis. We further provide a supplemental table with each participant’s total cumulative stress score for each study, which shows that participants varied in cumulative stress within and between age groups (S5 Table).”

[lines 379-381] “…95% CI: -2.45 to -0.29. No other comparisons were statistically significant. Table 3d provides means and standard errors by age group by stress group for each study and shows no overlap between high and low stress groups within age group for any studies.”

Other notes follow:

2. Lines 416-423: Indeed, the lack of interaction (whether computed by means of bayesian or frequentist meta-analysis) seems to confirm that ageing and stress may independently impair WM performance. However, I am concerned about the appropriateness of describing what amounts to an interaction effect (lines 426-427) after having shown that there is no interaction effect. I would limit the discussion of the effect on stress on young adults to the previous paragraph (around lines 412-413), expanding by referencing works that investigate the effects of stress on young adults cognition.

Alternatively, the same topic (effect of stress on YA) could be discussed around lines 444-446, when the topic is brought up again.

Author reply: We appreciate this feedback and have now deleted these lines (previously lines 426-427). We have added findings showing the effects of life events stress on YAs (lines 493-500) in lieu of this.

The changes in the manuscript read as follows:

[lines 493-500] “…memory, executive control and task shifting, explaining ≤ 58% of age-related variance. In light of such evidence, one may speculate that YA performance may better represent the impact of cumulative stress on cognition without the added variance contributed by ageing effects on performance. Indeed, studies of YAs that have investigated the effects of recent life events stress (4 to 6 months) found that high stress YAs showed blunted autonomic responsivity (95) and had poorer academic performance, psychological and physical health (96) than low stress YA. Low stress YAs were also better able to avoid risks when making decisions (97).Given that we evaluated a YA sample and their results also fell within...”

3. Furthermore, line 415: ... the evidence breached the insensitive range. Did the authors mean 'remained in the insensitivity range'?

Author reply: We have removed this sentence.

4. Finally, I believe the tables should be checked again for errors. The authors should either report all p values or mark the significant differences in the tables using formatting, though the first option would greatly be preferred. I find that reporting only the lowest p-value for each row creates a bit of confusion, and adds little information (knowing that p values are greater than .3 does not give more insight that knowing that the p values are greater than .05)

Author reply: Thank you, we have updated and/or reconfigured the tables as appropriate to improve clarity for the reader and have corrected the errors in Tables 3a-c (previously 1a-c) so that each age group has separately presented p-values.

5. In the same way, it is not clear why Mann-Whitney U tests were mentioned in the tables, but their values were not shown. It is possible that the authors chose to perform M-W U tests to check the results of a t-test, given possible deviations from t-test assumptions. However, deviations from normality do not imply that M-W U should be used (Fay, M.P. and Proschan, M.A. (2010)), and performing multiple tests does not lower the probability of type I errors. On the other hand, the authors did not specify whether Welch t-tests were used (this is preferred over performing a test for equality of variances followed by a t-test - Fay And Proschnan, 2010). If the choice of M-W U tests was on theoretical grounds, they should be reported instead of t-tests, not as robustness checks. Furthermore, it is unclear whether M-W U tests are significant whenever t tests are (the phrasing "similar outcomes" would suggest they are, but the p values are not reported)

Author reply: We have removed the M-W U test references. We have now updated Tables 3a-c (previously 1a-c) with notation stating which tests were used for each comparison, including the Welch t-test.

6. Some typos or mispellings found in tables include the last cell of Table 1a, which has a typo (0.0.78). I would also rephrase 'Chi-Square test for independence (low vs high stress) were performed by age group (p-value represents Yates' Continuity Correction)' as 'Chi-Square tests for independence (with Yates' Correction) were performed for each age group'. In tables 2 to 4, Credibility Interval should maybe be spelled as 'credible interval'.

Furthermore, In tables 2, 3 and 4 Bayes Factors are reported. However, Table 4 reports the common thresholds used to support H1 and H0 (3 and 1/3), while Table and 2 3 do not, and do not mark BF lower than 1/3 or greater than 3.

Author reply: Tables 3a-c (previously 1a-c): we have corrected typos/misspellings and have updated the tables to indicate when we used Yates’ Continuity Correction. Tables 4 – 7 (previously tables 2-4): we have replaced ‘Credibility Interval’ with 'credible interval' and we have marked BFs lower than 1/3 or greater than 3 in all the tables. 

7. Finally, If reporting p values as suggested before, asterisks could be removed. However, a doubt remains, as it is not clear what are the significance tests used from table 2 onwards and marked with asterisks.

Author reply: All asterisks have been removed from the tables.

---

## [Decision Letter · Decision Letter 1]

13 Aug 2023

Do non-traumatic stressful life events and ageing negatively impact working memory performance and do they interact to further impair working memory performance?

PONE-D-22-27705R1

Dear Dr. Wallace,

We’re pleased to inform you that your manuscript has been judged scientifically suitable for publication and will be formally accepted for publication once it meets all outstanding technical requirements.

Kind regards,

Thiago P. Fernandes, PhD

Academic Editor

PLOS ONE

Additional Editor Comments (optional):

Reviewers' comments:

Reviewer's Responses to Questions

**Comments to the Author**

1. If the authors have adequately addressed your comments raised in a previous round of review and you feel that this manuscript is now acceptable for publication, you may indicate that here to bypass the “Comments to the Author” section, enter your conflict of interest statement in the “Confidential to Editor” section, and submit your "Accept" recommendation.

Reviewer #1: All comments have been addressed

2. Is the manuscript technically sound, and do the data support the conclusions?

Reviewer #1: Yes

3. Has the statistical analysis been performed appropriately and rigorously? 

Reviewer #1: Yes

4. Have the authors made all data underlying the findings in their manuscript fully available?

Reviewer #1: Yes

5. Is the manuscript presented in an intelligible fashion and written in standard English?

Reviewer #1: Yes

6. Review Comments to the Author

Reviewer #1: The authors corrected typing errors, deepened the required parts and added what was suggested by the reviewers.

As a result, the article is now more precise and more complete.

7. PLOS authors have the option to publish the peer review history of their article (what does this mean?). If published, this will include your full peer review and any attached files.

Reviewer #1: No

---

## [Editor Report · Acceptance letter]

30 Aug 2023

PONE-D-22-27705R1 

Do non-traumatic stressful life events and ageing negatively impact working memory performance and do they interact to further impair working memory performance? 

Dear Dr. Wallace:

I'm pleased to inform you that your manuscript has been deemed suitable for publication in PLOS ONE. Congratulations! Your manuscript is now with our production department. 

Kind regards, 

on behalf of

Dr. Thiago P. Fernandes 

Academic Editor

PLOS ONE